# Current Neuroethical Perspectives on Deep Brain Stimulation and Neuromodulation for Neuropsychiatric Disorders: A Scoping Review of the Past 10 Years

**DOI:** 10.3390/diseases13080262

**Published:** 2025-08-14

**Authors:** Jonathan Shaw, Sagar Pyreddy, Colton Rosendahl, Charles Lai, Emily Ton, Rustin Carter

**Affiliations:** 1School of Medicine, California University of Science and Medicine, Colton, CA 92324, USA; 2Psychiatry, Eisenhower Health, Rancho Mirage, CA 92270, USA

**Keywords:** neuroethics, neuromodulation, deep brain stimulation, transcranial magnetic stimulation, transcranial direct current stimulation, brain–computer interfaces, schizophrenia, tourette syndrome, parkinson disease, major depressive disorder

## Abstract

Background: The use of neuromodulation for the treatment of psychiatric disorders has become increasingly common, but this emerging treatment modality comes with ethical concerns. This scoping review aims to synthesize the neuroethical discourse from the past 10 years on the use of neurotechnologies for psychiatric conditions. Methods: A total of 4496 references were imported from PubMed, Embase, and Scopus. The inclusion criteria required a discussion of the neuroethics of neuromodulation and studies published between 2014 and 2024. Results: Of the 77 references, a majority discussed ethical concerns of patient autonomy and informed consent for neuromodulation, with neurotechnologies being increasingly seen as autonomy enablers. Concepts of changes in patient identity and personality, especially after deep brain stimulation, were also discussed extensively. The risks and benefits of neurotechnologies were also compared, with deep brain stimulation being seen as the riskiest but also possessing the highest efficacy. Concerns about equitable access and justice were raised regarding the rise of private transcranial magnetic stimulation clinics and the current experimental status of deep brain stimulation. Conclusions: Neuroethics discourse, particularly for deep brain stimulation, has continued to focus on how post-intervention changes in personality and behavior influence patient identity. Multiple conceptual frameworks have been proposed, though each faces critiques for addressing only parts of this complex phenomenon, prompting calls for pluralistic models. Emerging technologies, especially those involving artificial intelligence through brain computer interfaces, add new dimensions to this debate by raising concerns about neuroprivacy and legal responsibility for actions, further blurring the lines for defining personal identity.

## 1. Introduction

Neuromodulation involves influencing the function of neurons through various modalities such as deep brain stimulation (DBS), transcranial magnetic stimulation (TMS), transcranial direct current stimulation (tDCS), electroconvulsive therapy (ECT), and brain–computer interfaces (BCI) [1]. The first use of neuromodulation techniques to treat psychiatric and neurological conditions dates back to the 19th century, where in 1804, Italian physicist Giovanni Aldini successfully treated melancholic (i.e., a term encompassing conditions now understood as various forms of depression) patients using electric stimulation [2]. Similarly, in 1886, a combined treatment modality involving tDCS and muscle faradization was used to rehabilitate motor function in chronic stroke patients [2]. Later developments in the 1930s saw the use of psychosurgical techniques like the leucotomy (a.k.a. lobotomy) and neuromodulation techniques like ECT being employed for a wide variety of psychiatric conditions such as schizophrenia, depression, and obsessive-compulsive disorder [2,3]. However, a disadvantage of the leucotomy, capsulotomy, cingulotomy, and other ablative neurosurgical procedures was their permanent effects on patients [3]. If a patient were to develop severe or otherwise intolerable side effects from an ablative neurosurgical intervention, surgeons could not reverse the procedure to restore prior function to the targeted tissue. This stands in contrast with DBS, a neurosurgical intervention that is also considered a neuromodulation technique, which is generally seen as reversible in contrast to the known irreversible effects of neuroablation [3,4]. With its first experiments occurring in the 1950s, DBS has only recently started to be employed for psychiatric causes like treatment-resistant schizophrenia or treatment-resistant depression [4]. Historically, DBS has primarily been used to treat neurologic disorders such as Parkinson’s disease, with recent retrospective studies indicating that DBS use in patients with Parkinson’s disease may lower their risk for mild cognitive impairments and falls [1,5].

In recent decades, there has been an increasing trend toward using these various treatment modalities for neuropsychiatric disorders [1]. ECT was initially approved in 1976 by the United States Food and Drug Administration (FDA) as a Class III device for use in psychiatric disorders such as schizophrenia, bipolar manic states, catatonia, or severe major depressive episodes associated with major depressive disorder or bipolar disorder [6,7]. Since then, ECT has been reclassified as a Class II device for patients requiring a rapid treatment response due to the severity of their psychiatric or medical condition [6,7]. Other neuromodulation therapies have subsequently been approved by the FDA for treating psychiatric disorders (e.g., TMS) or have received limited approval for certain conditions, remaining experimental for others (e.g., DBS) [7,8,9].

Depending on the neuropsychiatric disorder being treated, numerous areas of the brain have been proposed as targets for neuromodulation [3,10]. The medial forebrain bundle, which includes the ascending and descending white matter fibers connecting the ventral tegmental area with the nucleus accumbens, is proposed to play a role in reward processing and is an increasingly studied target for DBS for treatment-resistant OCD [3]. Some have conceptualized the brain as a Helmholtz machine that implements predictive coding to continuously generate and refine multiple predictions of the world through active sensorimotor exploration of the environment, with reward feedback systems guiding the overall process [10]. This reward feedback system is thought to include the nucleus accumbens, medial dorsal nucleus of the thalamus, pregenual anterior cingulate cortex/ventral medial prefrontal cortex, insula, and amygdala [10]. Through models such as the Helmholtz machine, various targets that were thought to inhibit autonomy have been proposed for treating OCD and addiction; said targets include the dorsal anterior cingulate cortex, nucleus accumbens, and/or the anterior limb of the internal capsule [10].

As the field of neuromodulation and psychosurgery has advanced, numerous ethical concerns have been raised due to its increasing prevalence in treatment-resistant neuropsychiatric disorders. Issues such as privacy of neural data and artificial intelligence directly interfacing with the human brain represent some of the more unique and modern ethical challenges addressed by neuroethics [11,12]. However, there have also been extensive discussions on the impact of neurotechnologies on patient identity, the risks and benefits of these therapies, the ethics of administering these treatments to severely ill patients who are either unable or unwilling to provide consent, and equitable access to these treatments [13,14]. This scoping review aims to synthesize the current discourse on the neuroethics of neuromodulation over the past decade (2014–2024) and to compare the ethical concerns associated with individual neurotechnologies. Upon reading this scoping review, clinicians will have a foundational understanding of the prevalent neuroethical issues associated with various neuromodulation techniques. The knowledge presented in this review will assist in clinical decision-making and facilitate more comprehensive explanations to patients who inquire about these experimental and emerging technologies.

## 2. Methods

### 2.1. Search Strategy and Study Eligibility

Per the PRISMA scoping review guidelines, 4496 references were imported from PubMed, Embase, and Scopus on 26 November 2024 using the following search terms: (ethics OR ethical) AND (neuromodulation OR (deep AND brain AND stimulation) OR DBS) AND (psychiatric OR neuropsychiatric OR Tourette OR Tourette’s OR depression OR anxiety OR schizophrenia OR personality). Studies met the inclusion criteria if they were published between 2014 and 2024 and explicitly discussed the ethics of neuromodulation for neuropsychiatric disorders. This study defines neuromodulation as any treatment that alters nervous system function (e.g., DBS, TMS, tDCS, BCIs) and neuropsychiatric disorders as those involving both psychiatric and motor or sensory components (e.g., schizophrenia with catatonia, Parkinson’s with mood symptoms). Three authors (J.S., S.R., C.R.) reviewed the imported references for eligibility, with two authors voting per reference and a third author resolving conflicts as needed. No protocol was registered to allow for flexibility in refining inclusion criteria post-extraction, given the theoretical nature of neuroethical literature and the goal of balancing inclusivity with conciseness.

### 2.2. Data Extraction

Two authors (J.S., C.L.) conducted the initial data extraction and summary, while two other authors (S.P., E.T.) conducted an independent data extraction and summary. Disagreements between these data extractions and summaries of findings were resolved by a separate author (C.R.).

## 3. Results

Of the 77 included in this scoping review, 65 were considered texts and opinions (literature reviews, commentaries, etc.), 10 were qualitative interview studies, and 2 were original articles with quantitative data. Details regarding reference selection are presented in Figure 1, while demographic information of the included references is presented in Table 1.

### 3.1. Personal Identity and Authenticity

The ethical and philosophical exploration of personal identity and authenticity has gained renewed relevance in light of emerging neuromodulation techniques, particularly with DBS [53,54]. These interventions, while therapeutically powerful, raise fundamental questions about what it means to remain “oneself” amid neurological transformation [28,53]. The existing literature primarily examines how DBS interacts with social constructs of identity, agency, and authenticity through a multitude of methods, ranging from empirical findings to conceptual frameworks [28,54,68]. This is due to the particularly invasive nature of DBS, which necessitates the surgical implantation of an electrode, compared to less invasive neuromodulation techniques such as TMS or ECT [48].

At the forefront of this debate are the qualitative interview studies and case reports that demonstrate changes in personality and symptom expression following DBS [3,16]. For instance, commentary on a case study of a patient who underwent DBS for severe Tourette’s syndrome highlighted how she reported an overall preferred outcome of a more “normal” self after a significant reduction of symptoms post-op, albeit with concerns from family regarding observed changes in her political and religious views, which may be perceived to have reflected neurologically induced changes in the patient’s intrinsic values [16]. Similarly, reviews of open-label studies on DBS for treatment-resistant depression have cited concerns regarding significant personality changes that lead to the questioning of self-identity [3]. These personal narratives underscore the emotionally complex and sometimes paradoxical consequences of DBS. However, the narrative that DBS significantly alters a patient’s personality, identity, agency, authenticity, autonomy, and self (PIAAAS) has been called into question by ethicists who posit that key neuroethics texts often cite a small number of empirical studies that have their conclusions misinterpreted as DBS affecting personality changes, rather than psychosocial reintegration difficulties [84]. Additionally, confounding factors, such as disease progression, medication adjustments, and premorbid psychiatric conditions, are not consistently accounted for as possible causes of personality changes in current meta-analyses of theoretical literature on the ethics and social consequences of DBS [84].

However, it also cannot be decisively concluded that patients undergoing neuromodulation do not experience a change in their sense of self, as there are reports that patients frequently struggle to evaluate whether their “improved” self post-DBS feels authentic or artificially induced [30,84]. Additionally, there are claims that prior literature has overemphasized the severity and prevalence of these changes [84]. Instead, the question of how authentic these changes in self/identity are and how one should view or define their identity has arisen, with various philosophical frameworks, such as narrative and relational identities, being presented in the literature [20,32].

Some patients expressed discomfort when their new behaviors and emotions seemed disconnected from their prior identity, raising concerns that DBS can disrupt one’s narrative identity and interfere with the authenticity of an individual’s perception of self [50,51]. Patients also expressed concerns about whether improvements from DBS are truly self-driven or the product of external machine manipulation [25]. Authors have also proposed a relational model that understands authenticity not as a static essence, but as a dynamic, socially mediated process [20,21]. This view is supported by a meta-analysis of qualitative interviews, which concluded that personal identity is maintained through narrative coherence, rather than psychological or biological consistency [24]. This was further supported in the literature with opinions that therapeutic interventions must align with the patient’s life story to be perceived as authentic [33]. It has also been noted that changes post-DBS in what patients deem important to their identity can disrupt the continuity of this narrative identity, potentially raising concerns about the validity of informed consent, as the self that provided the informed consent before DBS may not be the same self post-DBS [45]. This point was further elaborated in the literature by authors who stressed the importance of diachronic coherence, warning against preoccupation with momentary consent without consideration for long-term narrative identity [53,54,58].

Identity shifts have also been described through cognitive and psychological lenses, with researchers critiquing overly reductive models of the self and arguing that identity cannot be localized to neural substrates alone [38,81]. Instead, they propose an interactionist model in which cognitive, emotional, and social factors jointly construct identity [38,81]. This is supported by the argument that authenticity must be interpreted in light of an individual’s broader existential and contextual reality [27]. The importance of patient perceptions of their traits was demonstrated by a study that noted that even when depressive symptoms improve, patients sometimes describe their new behaviors as feeling artificial or imposed [4]. Similar findings have been documented in cases where therapeutic gains in obsessive-compulsive disorder (OCD) came with a subjective loss of continuity in self-experience [47,48]. These findings suggest that clinical success cannot be judged solely by symptom reduction and must also include the preservation of identity and authenticity, particularly in the context of protecting patient autonomy.

Social and cultural contexts are also critical in shaping experiences of identity change, as it has been found that cultural narratives about what constitutes a “normal” or “healthy” self influence how patients perceive the authenticity of their altered states [32,78]. Subsequent research has also noted that gender, ethnicity, and social status affect whether changes in personality or behavior are seen as empowering or disempowering [79]. This aligns with the belief in the literature that identity is never purely internal but rather emerges within a dynamic web of social expectations and values [77].

Several authors express caution about the potential medicalization or over-interpretation of identity changes [66,69,82]. Some warn against over-pathologizing normal fluctuations in personality post-DBS, and others argue that minor shifts in self-perception may be overblown in philosophical discourse, suggesting a need for measured, empirically grounded reflection [69,82]. There is also the opinion that identity changes should be expected to some degree in any major medical intervention and thus must be contextualized rather than sensationalized [66].

The importance of patient-centered frameworks has been emphasized in recent literature, with some authors proposing a relational autonomy model in which authentic decision-making is supported through meaningful dialogue and long-term reflection [56]. This is supported by studies that explored how narrative disruptions caused by DBS may be mitigated through therapeutic alliance and ethical guidance [72,73,74]. These studies suggest that clinicians should be trained not only in the technical aspects of DBS but also in the ethical and narrative dimensions of identity [72,73,74]. This is further elaborated on by researchers who suggest that while DBS can disrupt existing personality structures, it also offers opportunities for reconstruction and growth when integrated properly [67,75]. Religious or spiritual frameworks can sometimes facilitate this integration, offering patients interpretive tools to make sense of identity shifts [34]. In some cases, such frameworks can even enhance the sense of authenticity and purpose post-intervention [34].

Of note, multiple researchers have found that family members and caregivers often perceive identity changes differently from the patients themselves [16,35,43]. While patients may experience identity liberation secondary to the resolution of debilitating psychiatric symptoms, families often mourn the loss of familiar traits, raising ethical questions about whose perception of authenticity should guide treatment decisions [16,35,43]. This has led to some authors cautioning against framing identity in purely individualistic terms, advocating instead for a more inclusive, intersubjective approach [12].

In its current state, the literature paints a multifaceted picture of how neuromodulation interventions interact with personal identity and authenticity. While many patients experience therapeutic gains, these often come alongside existential questions about selfhood and autonomy. An integrative framework, specifically one that combines narrative, relational, ethical, and neurobiological insights, is needed to guide both clinical practice and equitable policy. Only by centering the lived experiences of patients in current discussions of neuromodulation ethics, and acknowledging the complexity of identity over time, can we ethically harness the transformative power of neuromodulation [20,21,57,73,84].

### 3.2. Autonomy and Informed Consent

Neuromodulation, particularly DBS and BCI, presents unique ethical challenges due to its capacity to alter cognition, emotion, and personality [11,31,79,81,82]. Central to medical ethics discourse are the principles of autonomy and informed consent, with scholars widely agreeing that the high-stakes and potentially identity-altering nature of surgical neuromodulation techniques in neuropsychiatric patients necessitates a reconceptualization of traditional informed consent frameworks [11,17,22,30,59,65]. However, that is not to say that lower-risk forms of neuromodulation like TMS are not discussed in the literature. TMS also continues to undergo clinical trials for the development of treatment protocols, and obtaining informed consent for these experimental treatments or off-label uses of TMS continue to be significant ethical concerns in the literature [23,52].

The principle of autonomy, as classically defined, centers on self-governance and freedom from coercion [31,33]. However, case studies have demonstrated shifts in personality post-DBS that raise questions about which version of the self had consented, thus introducing an extra dimension to informed consent for neuromodulation [16,73]. Similarly, others have highlighted the temporal dimensions of autonomy, arguing for respect for both pre- and post-DBS selves [58]. Although separate concepts from autonomy, one’s identity and the authenticity of one’s desires and values play an essential role in autonomous decision-making [16,61]. For instance, some researchers have discussed the impact of neurostimulation on authenticity, questioning whether post-DBS preferences can be considered truly autonomous [31]. Although neuromodulation can lessen the severity of neuropsychiatric symptoms and thus potentially improve patient agency, it can also induce unintended changes in mood, desires, and personality traits, which could alter a patient’s decision-making process in ways that they did not previously endorse or anticipate [31,46]. These issues highlight the difference between traditional autonomy, that which is measurable and typically employed in clinical settings, and what some deem as philosophical autonomy, which requires evaluative judgement and is not reducible to data alone [27]. There have been recommendations to adopt a relational model for assessing autonomy in neuromodulation, requiring continuous dialogue, reassessment, and the inclusion of family or caregivers in discussions to evaluate whether autonomy and agency are enhanced or diminished, rather than the more binary viewpoint of traditional autonomy [28,44,62,68,72].

Informed consent, while a legal and ethical prerequisite, is particularly fraught in DBS, as patients with psychiatric or neurological disorders may have impaired decision-making capacities, and many patients post-DBS across studies were found to have poor recall of key consent information [3,13,14,18,19,37,76,80]. For example, individuals with anorexia nervosa often demonstrate ambivalence toward recovery, denial of illness, and rigid thought patterns, which may compromise their ability to weigh the risks and benefits of neuromodulation effectively [25]. Moreover, a combination of the limited data on treatment outcomes, a tendency to overestimate benefits, and desperation due to severe or progressing symptoms makes it difficult to ensure adequate understanding when obtaining informed consent for DBS [22,23,39,41,50,65]. This can be further complicated when factoring in certain vulnerable patient populations, such as children, who may not always fully comprehend or have the ability to consent to complex neurosurgical procedures for neuropsychiatric conditions such as the commonly mentioned malignant Tourette syndrome [26,36,67].

To address this, some scholars propose procedural solutions to bolster informed consent. A prominent example is the Oxford Neuroethics Gold Standard Framework, which outlines safeguards for ethically responsible research and clinical applications for neurotechnologies like DBS [55]. Alternatively, the concept of “dynamic consent” is promoted as a model that allows for ongoing dialogue as a patient’s condition and self-concept evolve in the setting of an ongoing therapy [31,75]. This addresses concerns about disruptions of patient narrative identities post-DBS, allowing for improved tracking of a patient’s narrative identity and continued respect for their autonomy [10,30,31]. Similarly, there have been recommendations for iterative assessments of capacity and voluntariness throughout the DBS treatment journey [42].

A patient’s ability to refuse or withdraw from neuromodulation is essential to preserving patient autonomy; however, disorders of consciousness common across neuropsychiatric conditions may impair the patient’s capacity to provide informed consent [4]. Given the experimental nature of some neuromodulation techniques like DBS, patients can misunderstand the boundary between participation in a clinical trial versus receiving care, leading to therapeutic misconceptions [12,72,84]. This is in part due to feelings of desperation of patients and perceived “last resort” status that neuromodulation has for conditions like Parkinson’s disease or treatment-resistant depression, which can compromise a patient’s ability to evaluate risks and benefits objectively or potentially impair a patient’s ability to comprehend what neuromodulation is in the context of informed consent [18,26,34,38,50,65,70].

Due to the ethical complexity of obtaining informed consent for experimental neuromodulation treatments, there have been repeated calls for more patient-centered and ethically sensitive approaches for informed consent, which would involve increased transparency in not only surgical risks and benefits, but also the more metaphysical implications associated with neuromodulation [13,31,63,64,83]. Currently, the neuroethical literature on patient autonomy and informed consent primarily focuses on the agreement of the need to adopt more nuanced models for examining capacity and post-treatment autonomy.

### 3.3. Beneficence and Non-Maleficence

The principles of beneficence and non-maleficence are foundational in evaluating the ethical acceptability of neuromodulatory treatments, which can involve direct surgical intervention in the brain [11,13,36]. These principles demand that clinicians and researchers maximize therapeutic benefits (beneficence) while minimizing risks and harms (non-maleficence) [3,52,55]. The literature highlights several recurrent concerns in this regard, particularly in the use of DBS, TMS, tDCS, and BCIs for treating psychiatric conditions [11,13,15,23,29,36,52].

A particularly significant ethical concern involves the experimental status of many neuromodulatory techniques [3,11,23,26,49]. Currently, DBS remains investigational in most psychiatric contexts, including treatment-resistant depression, OCD, Tourette syndrome, addiction, and anorexia nervosa [18,25,26,36]. This experimental status and off-label usage of neuromodulation complicates risk–benefit evaluations and challenges the principle of non-maleficence by introducing conflicts of interest through the desire to collect research data or generate a profit or by causing therapeutic misconceptions due to limited clinical data [14,23,40,72]. Mixed clinical outcomes in randomized trials have raised questions about efficacy, patient selection, and stimulation parameters [3,4,18,50]. Many interventions proceed in the absence of rigorous empirical priors or animal model validation, leading to uncertain risk profiles that compromise informed assessments of benefit versus harm [19,48,64]. Questions of addiction, dependency, and behavioral overuse have also been raised with ethicists exploring how neuromodulation, particularly when targeting reward circuits for the treatment of addiction, may paradoxically create new forms of psychological reliance or pleasure-seeking behavior, requiring strict controls and ongoing ethical evaluation [77,78]. While such interventions hold promise for reducing harmful behaviors, they must be carefully monitored to prevent secondary harms resulting from overuse or motivational shifts [77,78].

Another major ethical concern exists for the use of neuromodulation in vulnerable populations, particularly in pediatric populations, that may benefit more from neuromodulation due to greater neuroplasticity, but due to limited clinical trials/data have poorly understood risks associated with neuromodulation as their brains continue to develop [11,15,26]. Studies on tDCS in children highlight potential benefits in disorders such as ADHD and autism, but they also underscore side effects and seizure risk, necessitating careful benefit–risk consideration [15]. Pediatric DBS introduces even more invasive risks, including infection, hardware complications, and psychological changes, with added concern about substitutive decision-making by parents who may be influenced by hope or desperation [26,40,50]. Providers have also reported concerns about DBS-related personality or mood changes in pediatric cases and their potential to disrupt authentic identity development [49,67].

Across age groups, identity alteration, emotional dysregulation, and loss of agency are highlighted as significant risks [48,51,57,63]. Even when neuromodulation alleviates clinical symptoms, it may inadvertently produce shifts in motivation, personality, or emotional tone that compromise patients’ sense of self [55,64,73]. Some ethicists stress the possibility of psychological harm or unintended personality changes, especially in populations like those with substance use disorders or severe psychiatric illness, where long-term data is lacking [14,48]. These changes may not be captured by conventional clinical metrics but hold considerable ethical weight, which requires multidisciplinary teams to assess and manage [50,67]. For instance, DBS has been reported to restore motor or affective functioning in ways that facilitate life goals [13] but also to produce unpredictable shifts in personal identity or social functioning, complicating the evaluation of benefit [51,57]. Some authors argue that beneficence in psychiatry cannot be measured solely through symptom relief but must incorporate subjective well-being, identity preservation, and meaningful quality of life [4,25].

Similarly, when attempting to practice non-maleficence, clinicians must factor in the concept of therapeutic misconception, wherein patients conflate research participation with guaranteed benefit or otherwise overestimate potential benefits while underestimating potential risks [48,72,73]. This further complicates assessments of risk and undermines informed consent, especially in the context of neuropsychiatric patients who may be desperate, face stigmatization, and suffer from significant illness-related impairments that may impair their capacity [72,76,80]. Moreover, profit-driven incentives and premature marketing may distort evidence bases and obscure the true risk–benefit profile of certain devices, especially in TMS for conditions like OCD [23,36]. Off-label uses of non-invasive neuromodulation, particularly TMS, pose similar concerns [23]. Though often perceived as safer than DBS, TMS carries risks of cognitive and mood side effects, especially in unregulated or commercially motivated contexts, highlighting the need for stronger ethical safeguards and professional standards [23,52]. Even ECT, although well-established, carries significant ethical concerns when used in patients lacking decision-making capacity [71]. While ECT may be life-saving in severe depression, this benefit must be cautiously balanced with the risk of cognitive impairment and the challenge of surrogate consent, requiring safeguards like ethics committee oversight and judicial review [71].

More controversially, the concept of applying neuromodulation to populations who may not perceive themselves as suffering, such as psychopaths or anorexia patients, has raised difficult questions about subjective benefit [25,37]. Some researchers challenge the ethical legitimacy of using DBS to modify behaviors in incarcerated psychopaths, arguing that these individuals do not report distress and may lack moral motivation, thereby undermining the standard justification for medical intervention [37]. Similarly, others describe cases where DBS improved Parkinsonian symptoms in pedophilic patients but led to hypersexual or disinhibited behavior, which introduces a significant ethical dilemma [46]. In such cases, ethicists advocate for improved informed consent processes with interdisciplinary oversight to ensure clinicians can balance a patient’s right to treatment with their ethical responsibility to prevent foreseeable harm to others [46].

Evidence quality remains a crucial consideration in fulfilling the principle of beneficence and facilitating a patient’s right to treatment, though clinical trial data quality varies wildly between neuromodulation modalities and is generally limited in sample size [14,26,46]. Compared to tDCS or TMS, DBS generally benefits from more mechanistic insight, controlled studies, and targeted neurosurgical precision, particularly in movement disorders and some psychiatric conditions like TS and TRD [29,36]. Likewise, some researchers challenge the framing of DBS as a strict “last resort,” noting that this paradigm may prolong patient suffering and obstruct timely, beneficial interventions [70]. In cases of malignant TS, where patients face high suicide risk and severe self-injury, DBS has demonstrated substantial efficacy (up to 96% improvement in some studies), leading authors to argue that delayed treatment risks violating non-maleficence through inaction [36]. Similarly, in dystonia or Parkinson’s disease, DBS has improved quality of life, employment capability, and social functioning, especially when introduced earlier in disease progression [13,50]. Despite these promising findings, clinicians must still take into account that the majority of this data comes from adult patients, limiting its generalizability to pediatric candidates for neuromodulation [26].

While treatment modalities like DBS appear to be supported by stronger and more consistent evidence in select conditions, particularly in movement disorders and in treatment-refractory cases of Tourette syndrome and OCD, it also raises serious ethical challenges concerning non-maleficence. The literature calls for enhanced risk stratification, transparent consent processes, longitudinal monitoring, and interdisciplinary ethical oversight to safeguard patient welfare [48,50,63]. Neuromodulation offers transformative possibilities, but beneficence must be contextualized to the patient’s circumstances and values, while non-maleficence must be upheld not only through minimizing harm but also by resisting premature application and safeguarding patients’ psychological integrity, autonomy, and dignity over time [55].

### 3.4. Justice

The literature reveals significant concerns regarding justice in the application of neuromodulation technologies for psychiatric disorders, particularly in terms of access, affordability, and equitable distribution of resources [18,34,42]. DBS and other neuromodulatory interventions such as ECT and TMS are typically high-cost treatments with limited insurance coverage, making them largely inaccessible to patients outside of affluent regions or socioeconomic classes [18,34]. Access to DBS remains highly centralized in specialized urban centers, creating geographic and institutional disparities that exclude many potential beneficiaries [12,50]. As a result, treatment opportunities are unevenly distributed, often privileging patients with the financial means and proximity to academic or research institutions [13,25].

One major structural barrier to access lies in the bureaucratic complexity of qualifying for neuromodulation, as patients are frequently required to exhaust multiple rounds of pharmacological and psychotherapeutic treatments and undergo invasive evaluations before qualifying for these advanced therapies, contributing to delayed care and prolonged suffering [22]. The experimental status of many neuromodulation technologies exacerbates this inequity, as off-label or investigational use of DBS or TMS is often not reimbursed and is primarily accessible only to wealthier individuals who can afford the high out-of-pocket costs associated with follow-up care and maintenance appointments [13]. This disparity is further underscored by the lack of continued access to DBS after the conclusion of clinical trials, which can result in significant ethical concerns when patients who benefit from the intervention are left without financial or medical support once research funding ends [42].

Commercialization trends pose additional threats to equitable care with the rise of private clinics offering neuromodulation therapies, particularly TMS. This introduces concerns about profit-driven practices, such as pressuring patients into excessive or unnecessary treatments, undermining informed consent and patient-centered care [22,23,52]. These practices may widen disparities by offering quicker access to those who can pay while neglecting comprehensive, evidence-based, and ethically sound care protocols [22,23,52]. Moreover, the dominance of a few companies in the neuromodulation market can distort scientific outcomes and introduce conflicts of interest that skew research and clinical application toward commercially viable models rather than those that serve broader public health needs [18]. Ethical oversight is often insufficient to counteract these trends, particularly when long-term psychosocial support and follow-up are neglected in favor of throughput and profitability [34,64].

Some researchers argue that the strict inclusion criteria of “pure” disease profiles for neuromodulation clinical trials limit access to these experimental treatments, excluding individuals with psychiatric comorbidities or pediatric patients who may arguably stand to benefit but are perceived to be at higher risk [12]. This selective inclusion reflects a broader tension between standardization and personalized care, often resulting in the marginalization of complex or vulnerable patients [12]. In legal and correctional contexts, neuromodulation interventions like DBS carry risks of coercive use, such as offering the procedure as a condition of parole, thereby blurring lines between treatment and social control [14].

TMS, often perceived as more accessible due to its non-invasive nature, still faces issues of cost and limited insurance coverage, particularly when used in off-label contexts such as PTSD or substance use disorders [13,23,52]. Without adequate public funding or geographic distribution of clinics, even this relatively safer neuromodulation treatment remains largely inaccessible to underserved populations [23,52]. As the demand for neuromodulation grows, justice-focused scholars and clinicians call for systemic reforms, including equitable access models, public funding, inclusion of patient voices in trial and treatment design, and holistic approaches that integrate these technologies into broader care systems [22,38,64].

There are calls for ethical frameworks that account for justice throughout the life cycle of neuromodulation treatments, from eligibility screening and trial participation to post-treatment support and long-term follow-up [72]. This includes developing regulatory mechanisms to ensure that commercially available neuromodulation technologies do not become tools of exclusion, coercion, or exploitation, and that they are implemented in ways that promote fairness, autonomy, and dignity for all patients, regardless of socioeconomic or geographic background [63,72]. Overall, while neuromodulation offers powerful tools for treating psychiatric illness, its ethical application is currently hindered by systemic inequities that must be addressed to realize the full promise of these interventions.

### 3.5. Privacy and Responsibility

The intersection of privacy and responsibility in the use of neuromodulation and neurotechnologies such as DBS and BCIs raises urgent and multifaceted ethical concerns [11,12,38,62]. A dominant theme across the literature is the challenge of protecting mental privacy in the face of continuous neural data collection, wireless communication, and the integration of computational feedback loops [11,38]. These technologies often involve real-time data streaming from the brain, leading to fears about unauthorized access, data breaches, or surveillance by third parties such as insurers, corporations, or governments [11,12,33,38,60]. This is especially problematic in pediatric and vulnerable populations, where long-term monitoring may result in lifetime exposure to neural data risks [11]. Scholars have increasingly called for legal and ethical recognition of “mental privacy” as a distinct right, one that encompasses protections beyond general data security laws [4,60].

A particularly acute concern centers on how neurotechnologies may affect “mental integrity,” defined as the capacity to maintain control over one’s thoughts, behaviors, and internal mental states [38,79,81,82]. Unlike traditional data privacy breaches, threats to mental integrity may arise from the very function of the technology itself. Closed-loop DBS systems or adaptive BCIs can subtly or overtly alter cognition, mood, or volition without the user’s full understanding or endorsement [12,81,82]. These influences may occur below the threshold of conscious awareness, thus compromising the user’s autonomy and potentially creating a mismatch between subjective experience and actual control [62,79]. These risks are amplified in the context of “digital twins,” where real-time models of patients’ brains are constructed from extensive physiological and behavioral data to predict or simulate psychiatric states [60]. While potentially useful for personalized treatment, this approach risks reductionism, epistemic opacity, and surveillance, raising additional concerns about consent, algorithmic accountability, and the erosion of patient dignity [60].

Beyond privacy, the issue of responsibility in the era of implanted neurotechnology has become increasingly complex [38]. Devices that influence behavior, cognition, or impulse control challenge conventional attributions of moral and legal responsibility [38]. For instance, if a neuromodulation device contributes to an individual committing a harmful act, either through malfunction or by unintended behavioral changes, accountability becomes ambiguous [38]. The presence of neurodata logs offers a form of auditable evidence yet also complicates questions of intentionality and authorship [4,38]. If a patient becomes emotionally or cognitively fused with their device, the boundary between person and technology blurs, challenging legal frameworks grounded in clear distinctions between internal agency and external instrumentation [38,62].

Additionally, concerns about coercion and social control have emerged, especially in criminal justice settings [12]. For example, undergoing DBS might be offered as a condition for parole or behavioral compliance, raising the specter of state-imposed manipulation and undermining the voluntariness of consent [12]. Such uses of neurotechnology could not only threaten privacy and integrity but may also distort or compromise the individual’s moral agency, leading to punitive or therapeutic interventions that blur ethical boundaries. In response to these concerns, multiple authors call for robust safeguards, including the development of neuro-specific privacy regulations, transparent design architectures, interdisciplinary ethical review boards, and consent procedures that go beyond acknowledging physical risk to fully disclose potential effects on thought, personality, and behavior [33,62,79,82]. They argue that ethical responsibility must be distributed across the entire system of stakeholders (e.g., designers, clinicians, manufacturers, regulators) and that protecting agency and mental privacy requires both technical and conceptual innovations [33,62,79,82].

Ultimately, the literature underscores that privacy and responsibility in neuromodulation are not merely extensions of existing medical ethics but domains requiring new frameworks and vigilance [33,62,79,82]. With the potential to access, modify, and record the most intimate elements of human experience, neurotechnologies demand an ethical infrastructure that treats mental privacy and autonomy as core, not peripheral, concerns [4,60,79].

## 4. Discussion

The discourse on the neuroethics of neuromodulation developed and evolved alongside advancements in the technologies it discusses, with the field generally agreeing on the status of certain ethical principles in neuromodulation, like neuroethicists’ concerns about justice [22,34,50]. However, other major principles of medical ethics, such as respect for patient autonomy, beneficence, and non-maleficence, have more varied perspectives due in part to the potential influence neuromodulation can have in shaping or completely changing a patient’s identity or sense of self [4,16,21,84]. This can be seen in the ongoing debates in the literature where there have been numerous frameworks proposed for defining and quantifying the concept of the self [16,20,24,34,47]. Conceptual frameworks such as “self-implant ambiguity” have been proposed for understanding how neuromodulation techniques like DBS can affect the self based on prior conceptual frameworks like John Sadler’s concept of self-illness ambiguity [21]. This framework comprises multiple forms of ambiguity: conceptual ambiguity, phenomenological ambiguity, and narrative or temporal ambiguity [21]. In conceptual ambiguity, the concept of the patient’s identity in relation to their condition is discussed, as patients may struggle with differentiating whether their symptoms or traits are part of their illness or themselves [21]. Meanwhile, phenomenological ambiguity has to do with a patient’s experience of mental states and whether their experiences (joy, irritability, etc.) are authentic or if they are induced by neuromodulation, particularly DBS [21]. Narrative or temporal ambiguity is similar to the established concept of narrative identity, with ambiguity arising when neuromodulation techniques disrupt one’s life story by abruptly or unexpectedly changing their circumstances by suddenly improving mood, drastically improving symptoms, or significantly altering behaviors and habits [12,21,32]. Other conceptual frameworks, such as the relational-narrative framework, have also been put forth, arguing that neuromodulation can impact both identity (who we are) and autonomy (how we govern ourselves) by altering a patient’s agency and competencies rather than through personality changes alone [32].

Early literature indicated that neurotechnologies like DBS could substantially change a patient’s personality, opinions, and habits in such a way that makes family members believe that they have become a different person [16]. Subsequent literature has disagreed with this interpretation, claiming that these personality changes are much rarer than the early literature suggests and that some of these changes are the authentic parts of the patient that had previously been masked by the symptoms of their neuropsychiatric disorder [16,84]. Additionally, it has been argued that key neuroethics texts often cite a small number of empirical studies but misinterpret their conclusions since these empirical studies often conclude psychosocial reintegration difficulties, not personality changes, following DBS [84]. The theoretical neuroethics literature has been critiqued for selectively quoting patients as stating, “I feel like an electric doll,” when this was a mistranslation and the patient had actually stated, “I am an electric doll,” which the critiquing authors note to be significant in terms of context [84]. Instead of conflating this quote into a philosophical issue regarding personal identity, the critiquing authors argue that the latter quote may involve a psychotic (delusional) episode, while the former could simply represent a playful and moody remark [84]. Despite the ongoing debate on the prevalence and importance of the impact of neuromodulation on personal identity, there is an understanding that these treatment modalities, particularly DBS, can significantly improve a patient’s symptoms and thus induce substantial changes in their circumstances, whether it be in lifestyle or their relationship with others [16]. This unresolved and controversial debate about the influence of neuromodulation on patient identity has its consequences, affecting the way clinicians and neuroethicists consider the major medical ethics principles of respect for patient autonomy through informed consent as well as beneficence and non-maleficence. There is a complex interplay between these ethical principles and patient identity, as these changes in patient identity can be seen as a benefit, a negative, or potentially both at the same time [16,20]. This further complicates obtaining informed consent, as what patients value or prioritize may significantly change post-neuromodulation, raising the question of whether the patient post-neuromodulation is the same patient that gave their consent [45].

For instance, there was the case of a “Ms. L” who underwent DBS for severe Tourette’s under a research protocol which not only resulted in significant reductions in the urge to tic 15 weeks post-op but also changes in her personality post-DBS, such as her change from being a shy, introvert to a notably more extroverted person [16]. Additionally, her family noted changes in her political and religious views, leading them to feel like she was no longer herself, and they requested device removal [16]. Ms. L stated that she felt “really great” and that she sees no problem with her changes post-DBS, preferring her post-DBS self [16]. The authors of Ms. L’s case considered the issue of whether DBS alters personal identity or intrinsic character [16]. One argument is that DBS has direct neuro-modification of the personality, but the authors felt this was unlikely due to the absence of other psychiatric side effects associated with the DBS target [16]. Another explanation for the change in personality/behaviors would be the liberation of the patient from debilitating symptoms [16]. This was deemed more likely as, with fewer tics, Ms. L no longer felt socially inhibited and could have discovered suppressed aspects of her personality [16]. Ms. L continued to demonstrate insight and self-awareness, suggesting that this was an authentic self-directed change rather than a psychiatric side-effect of DBS [16]. Although this improvement of symptoms can be positive for patients, there are also negative effects, such as the disruptions of roles for family members/caregivers (particularly spouses) who may struggle to adapt to the suddenly changed roles/changed personality of the patient [16]. These were seen in Parkinson’s DBS patients, where changes in personality have resulted in increased marital discord and even divorce [16]. While negatively perceived changes from DBS mandate intervention, positive transformation, if consistent with patient autonomy, should not be automatically pathologized or reversed [16]. Instead, providers should carefully consider the authenticity of these changes while weighing the risks and potential harm associated with device removal [16]. This is especially important because exaggerated claims of DBS risks may mislead patients and families, deter participation in beneficial therapies, and perpetuate stigma around psychiatric/neurological treatment [84]. Therefore, some neuroethicists believe that the field should move away from focusing on pathologized identity changes, and instead, clinicians and neuroethicists should be sensitive to identity-related disruptions, supporting ongoing narrative reconstruction instead of solely symptom management [21].

As it stands, the ethical and philosophical debates on the self will continue and become increasingly complex as neurotechnologies like closed-loop tDCS and the introduction of algorithms or artificial intelligence to BCIs become more prevalent. These burgeoning neurotechnologies have also raised new ethical concerns about neuroprivacy and the legal implications of actions taken by those with neuromodulation [12,79,82]. The literature on this topic was relatively limited in the 2014 to 2024 timeframe of this scoping review, but it is likely to be discussed significantly more in light of the increasing popularity of applying artificial intelligence to the medical field. Similarly, ethical concerns about financial and geographic barriers to neuromodulation treatments will also likely persist. As noted in the literature, more experimental techniques like DBS are restricted to academic centers, and patients will either need to partake in a clinical trial or pay out of pocket to maintain their implants [14,42,72]. TMS, although seen as less risky and less financially taxing, is still mostly available through private clinics, which may not accept the insurance of lower socioeconomic patients, thus also limiting its accessibility [52]. Despite these ethical concerns, neuromodulation continues to retain certain benefits compared to ablative psychosurgical techniques like the leucotomy, such as its reversibility [3]. DBS implants can be removed if side effects are intolerable or if the treatment is ineffective, while ablative procedures are permanent. This allows for neuromodulation treatment trials for patients that can be discontinued at will, similar to medication trials for psychotropic pharmaceuticals in psychiatric patients. Compared to psychotropic medications, neuromodulation procedures like DBS are higher risk due to the need for anesthesia and direct surgical intervention [3]. However, many neuromodulation treatments are considered a last resort for neuropsychiatric patients who have already failed or were unable to tolerate more conservative conventional therapies [22]. Therefore, neuromodulation techniques will likely continue to see continued research as a potential avenue of treatment for individuals with neuropsychiatric disorders that are treatment-resistant to psychotropic treatment options.

Regarding individual neurotechnologies in the existing literature, DBS appears to be discussed the most often, as it is seen as the most surgically risky; however, it also possesses the potential to have the most significant impact on alleviating patients’ neuropsychiatric symptoms [18,19]. Meanwhile, TMS is seen as relatively safer, though the risk of unforeseen or negative side effects remains [52]. The majority of the controversies with TMS appear to stem from its availability, which is increasingly provided through private clinics, as well as its off-label usage for various conditions [23,52]. With this in mind, the authors foresee increased discourse on the implications of artificial intelligence on neuromodulation and the self, as this is an ethical concern that existing frameworks do not adequately address. The differentiation between machine and the human self is a topic in which theoretical neuroethics will be most applicable, since quantitative studies will experience difficulties in defining and measuring such a philosophical state.

### Limitations

This scoping review is limited by its lack of formal analysis, as this was a review of mostly texts and opinions, rather than quantitative data, which is noted in the literature to be relatively scarce. Additionally, only 4 out of 77 references involved authors from countries/regions that are considered non-Western (Japan, Thailand, and Hong Kong), with the rest involving authors from countries considered Western (United States, Germany, Canada, etc.). This relatively low number of non-Western perspectives potentially results in a significant bias towards Western philosophical/ethical viewpoints, potentially due to the inclusion criteria of English full-text being a requirement. There may be other perspectives, informed by other cultural backgrounds and beliefs, that this scoping review did not include. The inclusion criterion of being published between the years of 2014 and 2024 also means that this scoping review did not include any newer references, such as those published in the year 2025, about artificial intelligence and neuromodulation. Future studies will need to create different cutoffs for publication dates in order to include these newer perspectives.

## 5. Conclusions

This scoping review has found that, over the past 10 years, the field of neuroethics has continued to debate and put forth conceptual frameworks for defining patient identity in the clinical context of neuromodulation, especially with the use of DBS. Many of these frameworks, such as that of self-implant ambiguity and the relational-narrative framework, address aspects of post-DBS changes in personality and behaviors in neuropsychiatric patients that may cause distress in patients or their families [16,84]. However, many of these frameworks are critiqued for only addressing certain aspects of post-DBS personality or identity changes [28]. To address this, neuroethicists have called for pluralistic, integrative models of the self, such as the “Pattern Theory of Self,” that are more holistic [28]. However, even these pluralistic frameworks may not be able to address the existential conundrum that is the fusion of human identity with artificial intelligence through BCIs. This has led to the rise of ethical concerns about neuroprivacy and the legal implications of actions taken by those with neuromodulation [12,79,82]. Given the rapid advancements in both neuromodulation and artificial intelligence technologies, this will likely be a topic neuroethicists will debate at length in the coming years. Other neuroethics topics, such as concerns about access to neuromodulation and standardized, ethical research remain relatively consistent, with neuroethicists broadly agreeing that there needs to be increased access to neuromodulation and that improvements to research protocols and reporting can be made.

## Figures and Tables

**Figure 1 diseases-13-00262-f001:**
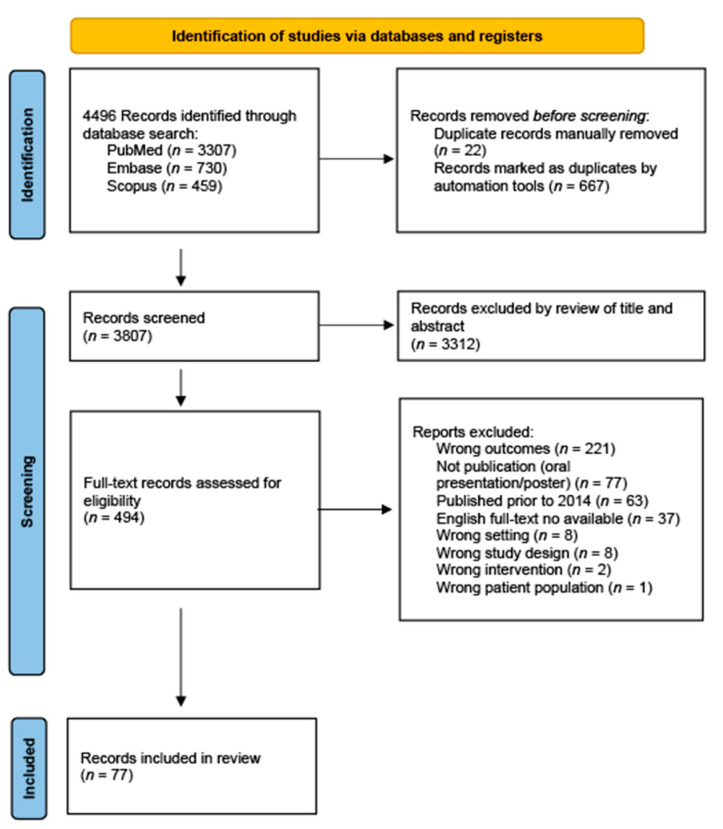
Reference identification and selection.

**Table 1 diseases-13-00262-t001:** Demographics of included ethics references.

Author	Year	Country	Neuroethic Topics
Auvichayapat [15]	2022	Thailand	Non-Maleficence
Beasley [16]	2016	United States	Personal Identity, Autonomy
Beeker et al. [17]	2017	Germany	Autonomy
Bergeron et al. [11]	2023	Canada	Autonomy, Beneficence
Bewernick et al. [18]	2022	Germany, United States	Autonomy, Justice, Beneficence
Bittlinger et al. [19]	2018	Germany	Autonomy, Maleficence
Bluhm et al. [20]	2020	United States	Personal Identity
Bluhm et al. [21]	2022	United States	Personal Identity
Cabrera et al. [22]	2022	United States	Justice, Autonomy
Cabrera et al. [23]	2024	United States	Non-Maleficence, Beneficence, Justice, Autonomy
Coman [24]	2017	Norway	Personal Identity, Authenticity
Dalton [25]	2021	United Kingdom	Non-Maleficence, Beneficence, Autonomy, Personal Identity
Davidson [26]	2018	Canada	Autonomy, Non-Maleficence, Beneficence
Delaloye [3]	2014	United States	Autonomy, Non-Maleficence, Beneficence
Deli [4]	2024	United Kingdom, United States	Non-Maleficence, Beneficence, Autonomy, Personal Identity, Privacy
Desmoulin-Canselier [12]	2020	France	Autonomy, Personal Identity, Privacy, Justice
Eijkholt et al. [13]	2017	United States	Non-Maleficence, Beneficence, Autonomy, Justice
Erler [27]	2021	Hong Kong	Autonomy, Personal Identity
Gallagher [28]	2021	United States	Autonomy, Personal Identity
Gault et al. [29]	2023	United States	Non-Maleficence, Beneficence
Gilbert [30]	2015	Australia	Personal Identity, Autonomy
Gilbert [30]	2021	United States, Australia, Switzerland	Personal Identity, Autonomy
Glannon [31]	2014	Canada	Autonomy
Goddard [32]	2017	Australia	Personal Identity, Autonomy
Gonzalez-Marquez [33]	2023	United Kingdom	Personal Identity, Justice
Grant [34]	2014	United States	Personal Identity, Justice, Autonomy
de Haan [35]	2017	Germany, Netherlands	Personal Identity
Hansen et al. [36]	2020	United States, Australia	Non-Maleficence, Beneficence, autonomy
Hubner et al. [37]	2016	Germany	Non-Maleficence, Beneficence, autonomy
Klein et al. [38]	2015	United States	Autonomy, Privacy, Personal Identity
Klein et al. [39]	2016	United States	Autonomy, Informed Consent
Kostick-Quenet et al. [40]	2023	United States	Beneficence, Non-Maleficence
Kubu [41]	2017	United States	Autonomy
Lazaro-Munoz et al. [42]	2018	United States	Justice
Lewis [43]	2014	Germany, United Kingdom, France, Canada	Personal Identity
Lo et al. [14]	2023	United States, Australia	Autonomy, Beneficence, Non-Maleficence, Justice, DBS
Mandarelli et al. [44]	2018	Italy	Informed Consent
Maslen et al. [45]	2015	United Kingdom	Autonomy, Personal Identity, DBS
Muller et al. [46]	2014	Germany, Switzerland	Beneficence, Non-Maleficence, Autonomy, DBS
Muller et al. [47]	2017	Germany	Personal Identity
Muller et al. [48]	2022	Germany, Belgium, Switzerland, Spain	Beneficence, Non-Maleficence, Personal Identity, DBS
Munoz et al. [49]	2021	United States	Autonomy, Informed Consent, Non-Maleficence, Beneficence, DBS
Munoz et al. [50]	2020a	United States	Pediatric DBS, Beneficence, Non-Maleficence, Justice, Personal Identity
Munoz et al. [51]	2020b	United States	Non-Maleficence, Beneficence, Autonomy, Justice, Personal Identity, DBS
Noda [52]	2024	Japan	Autonomy, Informed Consent, Beneficence, Non-Maleficence, Justice, Neuromodulation
Nyholm et al. [53]	2016	Netherlands	Personal Identity
Nyholm et al. [54]	2017	Netherlands	Personal Identity
Park et al. [55]	2017	United Kingdom	Informed Consent, Autonomy, Beneficence, Non-Maleficence, DBS
Pugh [56]	2020	United Kingdom	Personal Identity, Autonomy, DBS
Pugh et al. [57]	2017	United Kingdom	Personal Identity, Autonomy, Beneficence, DBS
Pugh et al. [58]	2018a	United Kingdom	Personal Identity, Autonomy, DBS
Pugh et al. [59]	2019	United Kingdom	Informed Consent
Rainey [60]	2022	United States	Privacy
Ridder et al. [10]	2016	New Zealand, United States, Germany	Autonomy
Roskies [61]	2015	United States	Autonomy
Schonau et al. [62]	2021	United States	Autonomy, Privacy, Responsibility
Shlobin et al. [63]	2020	United States	Beneficence, Non-Maleficence, Justice, Autonomy, Informed Consent
Shlobin et al. [64]	2022	United States	Beneficence, Non-Maleficence, Justice, Autonomy, Informed Consent
Siegel et al. [65]	2017	United States	Autonomy
Skorburg et al. [66]	2020	United States	Identity
Smeets et al. [67]	2018	Netherlands	Beneficence, Autonomy, Identity
Smith et al. [68]	2024	United States	Identity
Specker [69]	2019	United States	Identity
Stevens et al. [70]	2021	Australia	Beneficence, Autonomy
Takamiya et al. [71]	2021	Belgium, Japan	Beneficence
Thomson et al. [72]	2020	Australia	Autonomy, Non-Maleficence, Beneficence, Justice
Thomson et al. [73]	2021	Australia, Canada	Personal Identity, Non-Maleficence, Beneficence, Autonomy
Thomson et al. [74]	2023	Australia, Canada	Personal Identity, Autonomy
Unterrainer et al. [75]	2015	Germany	Autonomy, Informed Consent, Personal Identity, Authenticity
Versalovic et al. [76]	2023	United States	Autonomy, Informed Consent, Beneficence, Non-Maleficence
Viana et al. [77]	2019	Australia	Beneficence, Non-Maleficence, Personal Identity, Authenticity
Voigt [78]	2021	Germany	Beneficence, Non-Maleficence, Personal Identity, Authenticity
Warner et al. [79]	2023	United States	Autonomy, Informed Consent, Privacy, Responsibility, Personal Identity, Authenticity
Wexler et al. [80]	2021	United States	Autonomy, Informed Consent, Beneficence, Non-Maleficence
Witt et al. [81]	2017	Germany	Autonomy, Informed Consent, Privacy, Responsibility, Personal Identity, Authenticity
Zohny et al. [82]	2023	United Kingdom	Autonomy, Informed Consent, Privacy, Responsibility, Personal Identity, Authenticity
Zuk et al. [83]	2019	United States	Autonomy, Informed Consent

## Data Availability

No new data was created or analyzed in this review. Data sharing is not applicable to this review.

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
