# Peer review of "Current Neuroethical Perspectives on Deep Brain Stimulation and Neuromodulation for Neuropsychiatric Disorders: A Scoping Review of the Past 10 Years"

_diseases, 2025, doi:10.3390/diseases13080262_

Round 1
Reviewer 1 Report
Comments and Suggestions for Authors
Dear Authors
thank for the opportunity to review this manuscript.
This is an interesting and articulated study that explore the neuroethical discourse from the past 10 years on the 11
use of neurotechnologies for psychiatric conditions.
I think that it is a relevant topic for the journal and its readers and pertinent to many health care providers. Overall, this is an interesting topic and clinically relevant.
The title is appropriate.
The introduction shuold be revised to address the rationale and purpose for research question.
Furthermore, i suggest to add in introduction-line 36 this ref. (Deep brain stimulation in Parkinson's disease: A multicentric, long-term, observational pilot stud and y, by Scelzo, et al, 2019)
Paper is well presented, results, tables and figures are clear.
I feel that conclusions need to be improved.
Sentence between lines 259 and 261 is not claer, please rewrite it
For this reasons I decide to accept it, but there corrections should improve the quality of manuscript.
Author Response
The title is appropriate.
The introduction should be revised to address the rationale and purpose for research question. - The authors have revised the introduction to provide more extensive justification for the rationale of the scoping review and have clarified the purpose of the research question
Furthermore, I suggest to add in introduction-line 36 this ref. (Deep brain stimulation in Parkinson's disease: A multicentric, long-term, observational pilot stud and y, by Scelzo, et al, 2019) - Thank you for the reference, the authors have included this in the sentence for added context on the use of DBS in neurological disorders
Paper is well presented, results, tables and figures are clear.
I feel that conclusions need to be improved. - The authors have expanded on the conclusion
Sentence between lines 259 and 261 is not clear, please rewrite it - We have rewritten this sentence for better clarity
For this reasons I decide to accept it, but there corrections should improve the quality of manuscript.
Reviewer 2 Report
Comments and Suggestions for Authors
Very good and well written review.
Author Response
Very good and well written review.
Reviewer 3 Report
Comments and Suggestions for Authors
The authors present an insightful review on a significant topic in functional neurosurgery. However, several important points warrant consideration and revision to strengthen the manuscript:
Title. I recommend specifying in the title that DBS and neuromodulation are discussed in the context of psychiatric disorders.
Abstract. This key section of the paper requires revision. I assume that the 78 references mentioned in the results refer to studies that were evaluated following the application of inclusion and exclusion criteria. However, the conclusions are not entirely clear. While literature published since 2014 is referenced, the discussion includes studies predating 2014. It remains unclear whether the results and the cited literature adequately support the conclusions drawn.
Keywords. It would be beneficial to include a relevant keyword related to psychiatric disorders in this section.
Introduction. While modern neuromodulation has primarily been used for neurological disorders in recent years, it was initially more commonly applied to psychiatric conditions.
Methods. This section requires revision for better clarity and flow . It could be shortened, with some parts potentially could be integrated into the discussion section.
Discussion. This section could benefit from a stronger focus on the specific results of the review, rather than relying on overly general statements. An important distinction in relation to neuromodulation, as compared to ablative procedures, is that potential side effects of stimulation are typically reversible and can be adjusted.
Conclusions. The conclusions presented here seem somewhat different to the main points outlined in the abstract. In my view, this section could benefit from more specific and concrete statements, rather than remaining overly general.
Author Response
Title. I recommend specifying in the title that DBS and neuromodulation are discussed in the context of psychiatric disorders. - The title has been updated to contextualize the use of DBS and neuromodulation for neuropsychiatric disorders
Abstract. This key section of the paper requires revision. I assume that the 78 references mentioned in the results refer to studies that were evaluated following the application of inclusion and exclusion criteria. However, the conclusions are not entirely clear. While literature published since 2014 is referenced, the discussion includes studies predating 2014. It remains unclear whether the results and the cited literature adequately support the conclusions drawn. - Our apologies for the confusion, upon further review we noted that the essay by Wardrope was published in 2013 rather than 2014 (we had used Covidence for this scoping review, and it indicated that the essay was published in 2014). We have removed this reference as it is out of the acceptable publication timeframe for including in this scoping review. To our knowledge, that is the only reference that predates the 2014 cutoff for inclusion in this review. The PRISMA diagram, table, and all relevant sections have been updated to reflect this.
Keywords. It would be beneficial to include a relevant keyword related to psychiatric disorders in this section. - The relevant neuropsychiatric disorders have been added as keywords
Introduction. While modern neuromodulation has primarily been used for neurological disorders in recent years, it was initially more commonly applied to psychiatric conditions. - Thank you for pointing this out, the authors intended to discuss how DBS has historically primarily been used for neurological disorders. We have included new sentences detailing the history of various neuromodulation techniques.
Methods. This section requires revision for better clarity and flow . It could be shortened, with some parts potentially could be integrated into the discussion section. - We have edited the methods section for brevity and improved clarity.
Discussion. This section could benefit from a stronger focus on the specific results of the review, rather than relying on overly general statements. An important distinction in relation to neuromodulation, as compared to ablative procedures, is that potential side effects of stimulation are typically reversible and can be adjusted. - The authors have added more specific examples from the results of the review and have included a brief discussion about where neuromodulation stands in comparison to ablative procedures and psychotropic medications. The literature included in the review generally did not discuss ablative procedures, so our discussion on the topic is limited since the authors wish to include references that were included in the scoping review screening in the discussion section.
Conclusions. The conclusions presented here seem somewhat different to the main points outlined in the abstract. In my view, this section could benefit from more specific and concrete statements, rather than remaining overly general. - The authors have added more specific statements to the conclusion and reconciled/updated the conclusions in the abstract and the conclusion section
Reviewer 4 Report
Comments and Suggestions for Authors
This is a laudable attempt to address an ethical problem that dates back to the initial uses of psychosurgery in the 1950s. However, it lacks a comparison of DBS neuromodulation to lesioning and to medical management. Psychoactive medications used to treat these same neuropsychiatric conditions can be argued to do exactly the same things attributed here to DBS. DBS is adjustable and reversible, and so very like medication in this respect. This should be clearly distinguished from permanent lesioning (e.g., capsulotomy, cingulotomy, etc.) which confers permanent and irreversible changes. The choice of modality is a critical ethical issue. I would also suggest consolidating the discussion of what constitutes personal identity. While important to this topic, this is in large part a matter of philosophy as well as of neuroscience. The authors do a great job of thoroughly discussing this but at such length that it distracts from the main message of how neuromodulation can influence brain function. Concrete discussions of effects on measurable affective and cognitive parameters (and, as noted above, their reversibility) would be helpful.
Author Response
This is a laudable attempt to address an ethical problem that dates back to the initial uses of psychosurgery in the 1950s. However, it lacks a comparison of DBS neuromodulation to lesioning and to medical management. Psychoactive medications used to treat these same neuropsychiatric conditions can be argued to do exactly the same things attributed here to DBS. DBS is adjustable and reversible, and so very like medication in this respect. This should be clearly distinguished from permanent lesioning (e.g., capsulotomy, cingulotomy, etc.) which confers permanent and irreversible changes. The choice of modality is a critical ethical issue. - The authors have included a brief discussion of neuromodulation/DBS in relation to ablative procedures/psychotropic medication treatments in the discussion.
I would also suggest consolidating the discussion of what constitutes personal identity. While important to this topic, this is in large part a matter of philosophy as well as of neuroscience. The authors do a great job of thoroughly discussing this but at such length that it distracts from the main message of how neuromodulation can influence brain function. Concrete discussions of effects on measurable affective and cognitive parameters (and, as noted above, their reversibility) would be helpful. - The literature included in this scoping review was primarily texts and opinions, with many being philosophical in nature. This means that there are limited concrete numbers/statistics that the authors can include in the discussion, but we have attempted to remedy this by including specific examples from interviews with patients to highlight the potential effects of neuromodulation (mainly DBS).
Round 2
Reviewer 3 Report
Comments and Suggestions for Authors
The authors have thoroughly revised the paper, significantly improving it. The article is now ready to be accepted for publication in its current version.